# The prevalence of *M. tuberculosis* among acid fast bacilli cultures from military health system and veterans affairs beneficiaries in Hawaii and the Pacific Islands from 2002 to 2019

Elena M. Crecelius[1]*, Michael B. Lustik[2], Timothy S. Horseman[2], Milissa U. Jones[1,3]

1 Department of Pediatrics, Tripler Army Medical Center, Honolulu, Hawaii, United States of America,
2 Department of Clinical Investigation, Tripler Army Medical Center, Honolulu, Hawaii, United States of America, 3 Uniformed Services University of the Health Sciences, Bethesda, Maryland, United States of America

* elena.m.crecelius.mil@mail.mil

**Data Availability Statement:** The U.S. Department of the Army and the study investigators are committed to safeguarding the privacy of all

## Abstract

The prevalence of tuberculosis among military health system (MHS) and Veterans Affairs (VA) beneficiaries in Hawaii and the Pacific Islands has not been previously reported. Our analysis evaluates the prevalence of *M. tuberculosis* (MTB) among acid fast bacilli culture(s) (AFB) tested at Tripler Army Medical Center (TAMC) on Oahu, HI and describes demographic factors associated with positive samples. We analyzed 9,768 AFBs from 4,129 individuals with AFB specimens processed at TAMC from January 2002 to November 2019: of those who were tested 3,178 were MHS beneficiaries and 951 were VA beneficiaries. There were a total of 40 individuals with MTB-positive cultures over the period of study: 31 MHS beneficiaries and 9 VA beneficiaries. Of the MTB-positive specimens, 93% were from pulmonary samples while the remainder were from lymph node aspirates (5%) and peritoneal samples (2%). Cumulative incidence rates of MTB-isolation were 1.8 per 100,000 MHS beneficiaries and 1.2 per 100,000 VA beneficiaries, both of which were lower than reported incidence rates in Hawaii, the U.S.-affiliated Pacific Islands and the United States for the study period. MHS beneficiaries of Asian-Pacific Islander race or ethnicity had nearly 20 times higher odds of positive AFB than white MHS beneficiaries (OR = 19.56, 95% CI 5.52, 69.29, p = < 0.001). This study demonstrated a higher odds of MTB-positivity associated with Asian-Pacific Islander race or ethnicity and low incidence rates of TB among MHS and VA beneficiaries in Hawaii and the Pacific Islands when compared with the civilian population.

## Introduction

Tuberculosis is the leading infectious cause of death and one of the top ten causes of mortality worldwide [1]. Over the past century, rates of tuberculosis have declined in the United States, with the highest incidence rates reported in persons of Asian, Native Hawaiian or Pacific

military service members and their families. The study investigators have implemented measures to ensure participant anonymity is maintained in all reporting of research data. This allows for the publication of aggregate data only and specifically prohibits the publication or distribution of individual-level data. Distribution of de-identified participant-level data and accompanying research resources will require compliance with all applicable U.S. Department of the Army regulatory and ethical processes. Requests for these materials can be made to the Tripler Army Medical Center Department of Clinical Investigation via Dr. Zakariae Bram, PhD; Phone 808-433-6476; Zakariae.bram.mil@mail.mil.

**Funding:** The authors received no specific funding for this work.

**Competing interests:** The authors have declared that no competing interests exist.

Islander ethnicity or race [2–4]. Hawaii has the second highest case rate of TB in the U.S. with 8.4 cases per 100,000 persons in 2018, well above the national average of 2.8 cases per 100,000 persons [5]. Extremely high incidence rates of TB are also reported from the U.S.-affiliated Pacific Islands ranging from 3.9 to 338 cases per 100,000 persons in 2018 [6]. Over 80 percent of the individuals with reported tuberculosis in Hawaii were born in Asia or the U.S.-affiliated Pacific Islands [2, 5].

All branches of the U.S. military have active installations on the Hawaiian island of Oahu including the U.S. Army, U.S. Air Force, U.S. Navy, U.S. Marine Corps, and U.S. Coast Guard. The U.S. military health system serves active duty personnel, activated National Guard and Reserve Corps personnel, persons retired from active duty service, and dependents. Both military health system (MHS) beneficiaries and Veterans Affairs (VA) beneficiaries from Hawaii and the Pacific Islands receive care at Tripler Army Medical Center (TAMC), which is the sole military hospital on the Hawaiian island of Oahu.

The prevention, diagnosis and screening of TB are critically important to public health and the medical readiness of active duty U.S. military personnel. The nationally reported incidence of TB is lower among active duty personnel than the general population with fewer than one case per 100,000 persons [7, 8]. Mancuso and colleagues noted gaps in the reporting of TB cases to the Armed Forces Health Surveillance Center which were similar to civilian reporting rates and likely due to incomplete data availability with an unknown true positive rate [8]. U.S. military personnel are put at a higher risk of infection with TB during deployment or other service in endemic regions, when interacting with foreign personnel, civilians or detainees from TB endemic regions and while living in congregate settings [9]. While most cases of active TB in the military occur due to activation of latent infection, one study found that 24% were associated with deployment to endemic regions [7]. Screening regulations are in place to test all military personnel for tuberculosis after high-risk exposures or deployments [10]. There are limited data regarding the rates of tuberculosis in other military health beneficiaries such as dependents or retirees. Published reports of TB in these populations are limited to branch-specific data or case reports which may indicate a low overall prevalence of TB among this population [11–14].

Understanding the epidemiology of TB among military health system and Veterans Affairs beneficiaries is of particular importance in Hawaii and the Pacific Islands given the observed higher incidence rates of TB as compared to other parts of the United States. Presently, the prevalence of TB in MHS and VA beneficiaries in Hawaii and the Pacific Islands is unknown. This analysis evaluates the prevalence of *M. tuberculosis* among MHS and VA beneficiaries in Hawaii and the Pacific Islands who were tested for TB at Tripler Army Medical Center from January 2002 to November 2019 and describes factors associated with positive AFB cultures.

## Material and methods

### Study design and setting

This retrospective analysis evaluates the prevalence of *M. tuberculosis* in AFBs from 2002 through 2019 at TAMC on Oahu, HI. The TAMC diagnostic microbiology laboratory processes clinical samples from MHS and VA beneficiaries from Hawaii and the Pacific Islands. All AFB samples included in this study were submitted to the laboratory for processing during routine patient care with AFB testing ordered per physician discretion.

### Acid-fast bacilli culture sample processing

Clinical specimens from body fluids were processed using the NALC-NaOH method with subsequent AFB smear and culture performed. If growth was detected, molecular testing using

DNA probes for identification of MTB, *Mycobacterium avium complex*, and *Mycobacterium gordonae* was performed for confirmation at Tripler Army Medical Center [15, 16]. Positive samples or isolates were subsequently sent to reference laboratories for organism identification, confirmation, and antibiotic susceptibility testing, as this testing was unable to be performed at TAMC. Final results were recorded in the patient's electronic health record (EHR) as positive, including speciation and susceptibilities, or negative.

## MHS and VA beneficiaries

We refer to individuals eligible for healthcare within the military health system as MHS beneficiaries. For purposes of this analysis, MHS beneficiaries include active duty and activated National Guard and Reserves Corps personnel across all branches of service including the U.S. Army, U.S. Navy, U.S. Air Force, U.S. Marine Corps, and the U.S. Coast Guard. Additionally, persons retired from active duty service and dependents, defined as spouses, children and other eligible family members of active duty or retired personnel, were all considered MHS beneficiaries. MHS beneficiary enrollment data was obtained from the TAMC Business Operations Division Database for each year over the study period, as available; data was unavailable for the year 2002.

We refer to individuals eligible for healthcare within the Veterans Affairs system as VA beneficiaries. VA beneficiary enrollment data was obtained from the VA Pacific Islands Health Care System for each year over the study period, as available; data was unavailable for the years 2017–2018.

## Data collection and analysis

The results of AFBs were aggregated over the study period. Samples were excluded from analysis if the final results were not recorded by November 2019 or if they were obtained from individuals with MHS beneficiary status of Pacific Island Health Care Project, NATO, civilian or unknown. Samples with final results positive for *M. tuberculosis* were categorized as 'MTB-positive.' Samples that had a negative final result or that were positive for a non-tuberculous mycobacterial species were categorized as 'MTB-negative.' The first positive sample was included for individuals with multiple MTB-positive samples. The first negative sample was included for individuals with multiple MTB-negative samples. Individuals who had both positive and negative samples only had their first positive sample included. Demographic data belonging to each patient were collected from the electronic health record at the time of their included sample, including age in years, sex, race or ethnicity and military status. This study was deemed as exempt from IRB review by the Tripler Army Medical Center Human Research Protections Programs with the need for informed consent waived.

The incidence rates of MTB were determined by the total number of individuals with MTB-positive samples per total number enrolled individuals for both MHS and VA beneficiaries. Annual incidence rates were calculated using the number of new cases of TB per reported enrolled beneficiaries by year. Cumulative incidence rates were calculated using the total number of new cases over the study period per total number of reported enrolled beneficiaries over the study period. Chi-square tests and Fisher's exact tests were used to evaluate differences in demographic features between MTB-negative and MTB-positive individuals. Multivariable logistic regression models were used to estimate adjusted odds ratios for demographic factors associated with MTB-positive results and to estimate the odds of MTB-positive results by year. All analyses were conducted using SAS statistical software version 9.4 [SAS Institute, Cary, NC].

## Results

The demographic characteristics of the individuals included in this analysis are presented in Table 1. The majority of the MHS beneficiaries were male, of white or other/unknown race or ethnicity, and categorized as dependent status. The majority of the VA beneficiaries were male and over the age of 40 years. When comparing individuals with MTB-positive results to those with MTB-negative results, statistically significant differences were observed in MHS beneficiaries with regard to race or ethnicity.

A total of 9,768 specimens collected from 4,129 individuals resulted at TAMC for AFB smear with reflex to culture between 2002 and 2019. There were 31 MTB-positive results from 3,178 MHS beneficiaries and 9 MTB-positive results from 951 VA beneficiaries, yielding a cumulative 0.97% prevalence by person and 0.98% prevalence by specimen. A total of 22 individuals had more than 1 MTB-positive result. This study included only those samples performed at TAMC laboratory. Some of the follow-up testing was performed through the TAMC laboratory, however other individuals had their follow-up testing performed by the Department of Health in their local jurisdiction as part of their Directly Observed Therapy.

Between 2003 and 2019 the number of MHS beneficiaries in Hawaii ranged from 81,725 to 118,361; enrollment data for 2002 is unavailable. Between 2002 and 2019 the number of VA

**Table 1. Characteristics of Military Health System (MHS) and Veterans Affairs (VA) beneficiaries with acid-fast cultures processed at Tripler Army Medical Center, from January 2002 to November 2019.**

| Characteristics | Total MHS beneficiaries tested for AFB | MTB-positive MHS beneficiaries | p-value | Total VA Beneficiaries tested for AFB | MTB-positive VA beneficiaries | p-value |
|---|---|---|---|---|---|---|
| | n | n (%) | | n | n (%) | |
| **All Age (years)** | 3178 | 31 (1.0%) | 0.362 | 951 | 9 (0.9%) | 0.664 |
| 0–21 | 639 | 3 (0.5%) | | 1 | 0 (0.0%) | |
| 22–40 | 970 | 13 (1.3%) | | 66 | 0 (0.0%) | |
| 41–65 | 860 | 9 (1.0%) | | 428 | 3 (0.7%) | |
| >65 | 709 | 6 (0.8%) | | 456 | 6 (1.3%) | |
| **Sex** | | | 0.143 | | | 1.000 |
| Male | 1869 | 14 (0.7%) | | 916 | 9 (1.0%) | |
| Female | 1309 | 17 (1.3%) | | 35 | 0 (0.0%) | |
| **Race/Ethnicity** | | | <**0.001** | | | 0.091 |
| Asian-Pacific Islander | 639 | 21 (3.3%) | | 283 | 6 (2.1%) | |
| Black | 272 | 3 (1.1%) | | 62 | 0 (0%) | |
| White | 1240 | 3 (0.2%) | | 350 | 1 (0.3) | |
| Other/Unknown[a] | 1028 | 4 (0.4%) | | 256 | 2 (0.8%) | |
| **Military Status** | | | 0.056 | | | |
| Dependent[b] | 1655 | 17 (1.0%) | | - | - | |
| Retirees | 736 | 4 (0.5%) | | - | - | |
| Active Duty | 744 | 8 (1.1%) | | - | - | |
| National Guard/ Reserves Corps[c] | 43 | 2 (4.7%) | | - | - | |
| VA | - | - | | 951 | 9 (0.9%) | |

[a] 927 were reported as other, 274 were reported as null, 76 were reported as unknown, and 6 were reported as Western Hemisphere Indian.

[b] Dependent includes spouses, children and other family members of active duty persons.

[c] Refers to National Guard/Reserves Corps who were activated, thus eligible for military healthcare.

Statistically significant p-values (p < .05) are in bold. p-values are based on chi-square tests for age, race/ethnicity, and MHS beneficiary status, and Fisher's exact test for sex.

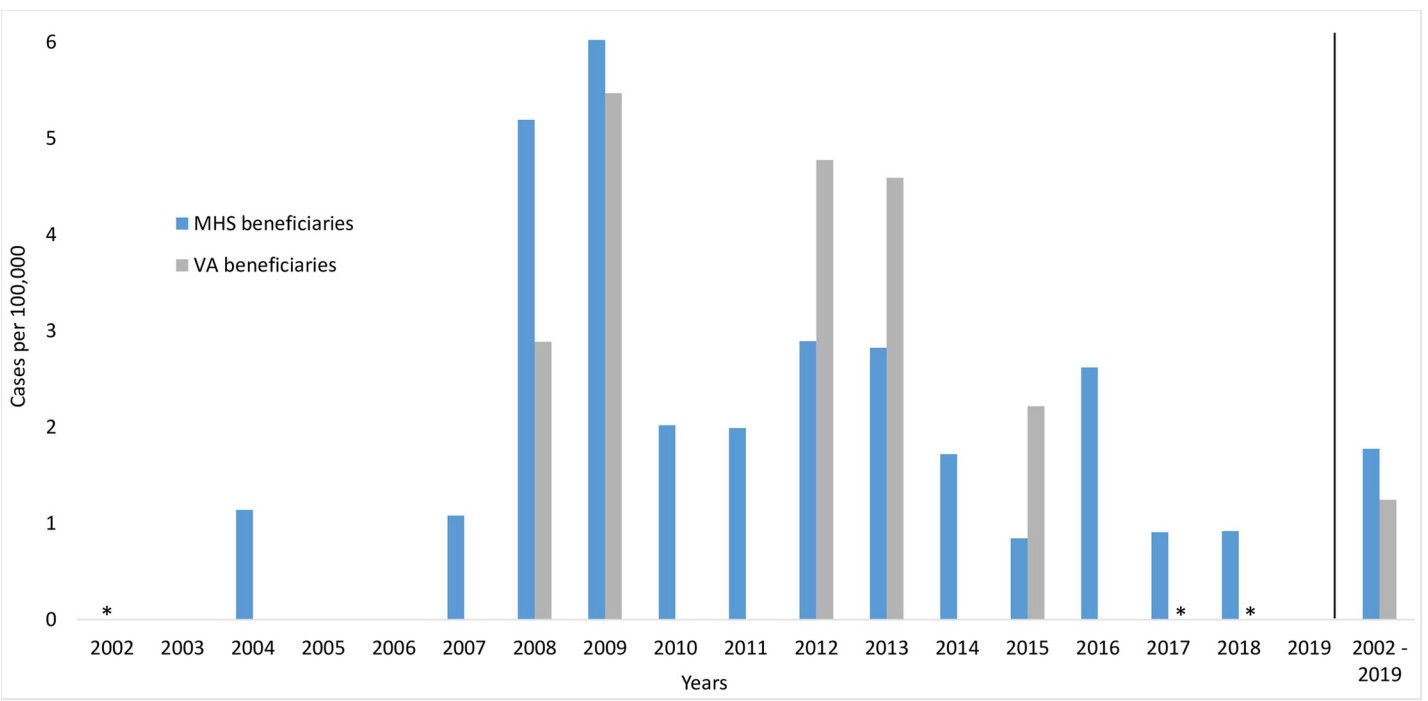

**Fig 1. Annual and cumulative incidence rates of MTB among Military Health System (MHS) and Veterans Affairs (VA) beneficiaries in Hawaii and Pacific Islands, 2002–2019.** * No data available for 2002 for MHS beneficiary population; no data available for 2017, 2018 for VA beneficiary population.

beneficiaries in Hawaii and the Pacific Islands ranged from 27,962 to 54,850; data for 2017–2018 is unavailable.

The cumulative case rate of MTB for our study period (2002–2019) for MHS beneficiaries was 1.8 cases per 100,000 persons (Fig 1). The annual case rate for MHS beneficiaries for 2002 was not included as the total number of MHS beneficiaries in Hawaii was not available for that year. The cumulative case rate of MTB for our study period (2002–2019) for VA beneficiaries was 1.2 cases per 100,000 persons (Fig 1). The annual case rate for VA beneficiaries for 20016 and 2017 was not included as the total number of VA beneficiaries in Hawaii and the Pacific Islands were not available for those years. Among MHS beneficiaries, the proportion of positive MTB cultures was significantly higher during the years 2007 to 2009 (12/440, 2.7%) as compared to 2002–2006 (1/684, 0.1%, p<0.001) and 2010–2019 (18/2054, 0.9%, p = 0.003). The trend was the same among VA beneficiaries, but results were not significant due to smaller sample sizes (2.4% vs. 0.0% for 2007–2009 vs. 2002–2006, p-0.076; and 2.4% vs 0.9% for 2007–2009 vs 2010–2019, p = 0.168).

The sources of all specimens were: abscess, blood, body fluid, bone marrow, cerebrospinal fluid, gastric aspirate, lymph node aspirate, other (unspecified), pericardial fluid, peritoneal fluid, pulmonary, stool, synovial fluid, tissue, urine, and wound. Of the MTB-positive samples (n = 105) 93% came from pulmonary sources with 5% from lymph node aspirates and 2% from peritoneal specimens (Fig 2A). Of the MTB-positive pulmonary samples (n = 98) a majority came from sputa (70%), while the remainder came from bronchoalveolar lavage (15%), pleural fluid (9%), bronchial washes (5%), and tracheal aspirates (1%) (Fig 2B).

Among the MTB-positive isolates, there was only one sample with drug resistance to isoniazid, the remainder were pan-susceptible on laboratory testing. The rate of isoniazid-resistant MTB in our study was 2.5% (1 of 40). Of note, there were no multidrug-resistant TB (MDR TB) or extensively drug-resistant TB (XDR TB) isolates.

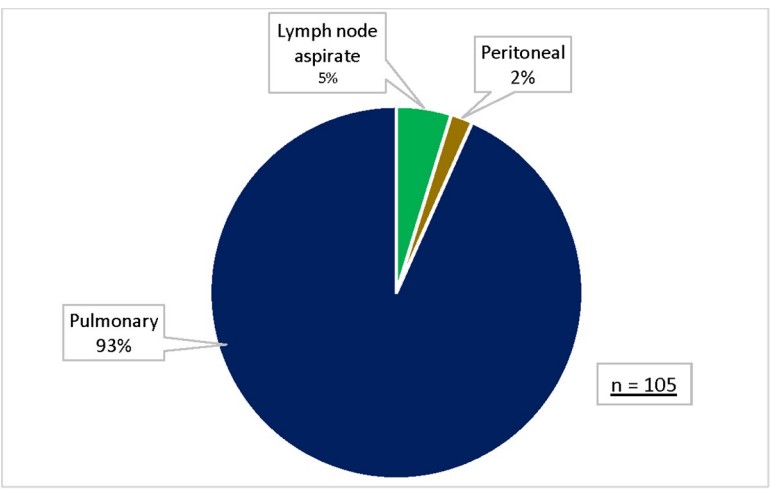

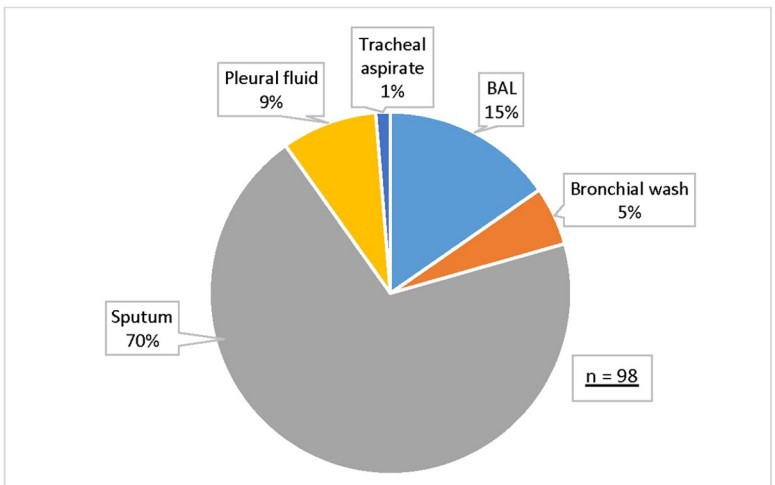

**Fig 2. Source of MTB-positive specimens from Military Health System (MHS) and Veterans Affairs (VA) beneficiaries processed at Tripler Army Medical Center, 2002–2019.** A. All MTB-positive specimens B. MTB-positive pulmonary specimens. Sources of all specimens: abscess, blood, body fluid, bone marrow, cerebrospinal fluid, gastric aspirate, lymph node aspirate, other (unspecified), pericardial fluid, peritoneal fluid, pulmonary, stool, synovial fluid, tissue, urine, and wound.

After controlling for gender, age, and military status, Asian-Pacific Islander MHS beneficiaries had nearly 20 times higher odds of MTB-positive results than whites (OR = 19.56, 95% CI 5.52, 69.29, p = < 0.001). There was no significant association between other demographic factors and MTB-positivity (Table 2).

## Discussion

We demonstrate a low yield of MTB among clinical specimens recovered from MHS and VA beneficiaries from Hawaii and the Pacific Islands processed at Tripler Army Medical Center from 2002 to 2019. Overall we observed less than 1% positivity among all specimens received in the study period in comparison to previously reported civilian prevalence data in the state of Hawaii which indicated prevalence of almost 2% among laboratory specimens [17] There were 0–6.0 (cumulative 1.8) cases per 100,000 MHS beneficiaries and 0–5.5 (cumulative 1.2) cases per 100,000 VA beneficiaries during the study period which are lower than the incidence

**Table 2. Characteristics of Military Health System (MHS) and Veterans Affairs (VA) beneficiaries associated with MTB-positive specimens, Tripler Army Medical Center laboratory, 2002–2019.**

| Characteristic | MHS Beneficiaries | p-value | VA Beneficiaries | p-value |
|---|---|---|---|---|
| | OR (95% CI^) | | OR (95% CI^) | |
| **Age** | | | | |
| 0–21 | Referent | | Referent[c] | |
| 22–40 | 1.50 (0.38–5.94) | 0.565 | Referent[c] | |
| 41–65 | 1.01 (0.26–3.90) | 0.986 | Referent[c] | |
| >65 | 0.70 (0.16–2.95) | 0.622 | 2.03 (0.50–8.21) | 0.320 |
| **Sex** | | | | |
| Male | Referent | | Referent | |
| Female | 1.77 (0.78–4.01) | 0.171 | Not estimable[d] | |
| **Race/Ethnicity** | | | | |
| White | Referent | | Referent | |
| Asian-Pacific Islander | **19.56 (5.52–69.29)** | **<0.001** | 7.28 (0.87–60.86) | 0.067 |
| Black | 3.46 (0.69–17.42) | 0.132 | Not estimable[d] | |
| Other/unknown | 1.42 (0.32–6.38) | 0.646 | 2.73 (0.25–30.27) | 0.414 |
| **Military Status** | | | | |
| Dependent/Retiree[a] | Referent | | | |
| Active Duty/National Guard/Reserves Corps[b] | 3.06 (0.97–9.64) | 0.056 | | |

^ CI = confidence interval.

[a] Dependent includes spouses, children and other family members of active duty persons or eligible retirees.

[b] National Guard and Reserves Corps activated, thus eligible for military health care.

[c] VA beneficiaries 0–65 years of age combined due to small sample sizes and lack of MTB-positive samples among this population.

[d] Odds ratios for females and individuals of black race/ethnicity not estimable due to lack of MTB-positive samples among these groups.

Logistic regression with 95% confidence intervals were used to calculate adjusted odds ratios. Statistically significant odds ratios and p-values (p<0.05) are in bold.

rates of tuberculosis in Hawaii but similar to the national incidence rates during the study period. Each year over 100 cases of tuberculosis are reported to the Hawaii Department of Health (HDH) yielding a case rate around 8.2 to 11.9 per 100,000 persons over the same time period [2]. The estimated case rate in our population was similar to the average national case rate of 2.7 to 5.2 cases per 100,000 persons over the study period [18]. Additionally, we observed that 2.5% of MTB-positive isolates recovered were resistant to isoniazid. Over similar years (2006–2015), the HDH reported the prevalence of TB drug resistance to isoniazid between 1% to 22% and MDR-TB between 0% to 7% [2]. The reason for the observed differences between the incidence in our study population compared with the civilian population remains unclear and is worthy of future analysis.

The incidence rates of positive AFBs varied by year for reasons that have not been elucidated (Fig 1). There were no identifiable changes in testing protocol, laboratory availability or other institutional reason across the years to explain the significant difference in annual prevalence. Additionally, the case rate of TB in the state of Hawaii was stable across the same time period [2]. Of note, Mancuso et al. reported the highest rate of tuberculosis cases in active duty U.S. military personnel in 2009 [7]. Only one of the 7 individuals in our study with MTB-positive results in 2009 were active duty personnel. These analyses could not account for why the highest incidence was found in 2009 in our study.

After controlling for other factors, MHS beneficiaries of Asian-Pacific Islander race or ethnicity had nearly 20 times higher odds of MTB-positive clinical samples than other races or

ethnicities included in this analysis. Our findings are consistent with well-established data which demonstrate a higher prevalence of TB among Asian-Pacific Islanders, Native Hawaiian and other Pacific Islander minorities [2–4]. Of note, a 2013 study by Mancuso et al. that examined surveillance trends of laboratory-confirmed TB cases in U.S. armed forces and risk factors associated with tuberculosis from 1998 to 2012, also found the highest incidence rate of TB among Asian-Pacific Islanders [7]. A 2017 study of MTB-positive AFB results from patients in the Kaiser-Permanente Health system in Hawaii between 2005 and 2013 found the lowest TB rates among Native Hawaiian, Pacific Islanders, Japanese and white persons and the highest rates among persons of Filipino, Korean and Vietnamese ethnicities [17]. Our study did not differentiate the different countries of birth of the individuals with MTB-positive specimens.

Future analyses derived from this study will focus on elucidating the key sociodemographic factors present in the 40 individuals with MTB-positive cultures to determine if there are certain factors which place MHS and VA beneficiaries at greater risk of TB.

## Strengths and limitations

This is the first study evaluating the laboratory epidemiology of tuberculosis among the MHS and VA beneficiary population served at TAMC. We present results from MTB clinical specimens spanning a 17+ year time period and include thousands of samples. Our study only included MHS and VA beneficiaries with AFBs processed within the TAMC laboratory system, thus cannot speak to the totality of MTB among MHS and VA beneficiaries present in Hawaii and the Pacific Islands. The demographic data included in this analysis was limited to what was reported in the EHR. While likely reliable for age and gender, race or ethnicity status may have been incorrectly recorded or omitted which contributes to the large number of "other/unknown" (1,028 MHS beneficiaries and 256 VA beneficiaries, 32.3% and 26.9% of study population, respectively) seen in this analysis. Our study focused on isolation of MTB from clinical samples rather than clinically diagnosed TB disease. It is plausible that certain MHS beneficiaries with true TB disease never had samples processed in our system, thus we could have underestimated the prevalence of TB in our population. We suspect this scenario to be exceedingly rare as use of AFB is paramount to managing tuberculosis, particularly with regard to evaluation for resistant MTB.

## Conclusion

We observed a low prevalence of *M. tuberculosis* among AFBs derived from MHS and VA beneficiaries from Hawaii and the Pacific Islands from January 2002 to November 2019. Our findings contrast the high prevalence of *M. tuberculosis* among the civilian population of Hawaii and the Pacific Islands during this time period. Our results corroborate civilian data showing an association between Asian-Pacific Islander race or ethnicity and MTB-positive results.

## Acknowledgments

We would like to acknowledge Chang, Kaulana and Membrere, Nemy from Business Operations Division, Tripler Army Medical Center for providing data on MHS beneficiary enrollment as well as Hinds, Sean from the VA Pacific Islands Health Care System for providing data on VA enrollment.

## Author Contributions

**Conceptualization:** Elena M. Crecelius, Timothy S. Horseman, Milissa U. Jones.

**Data curation:** Elena M. Crecelius, Michael B. Lustik, Timothy S. Horseman, Milissa U. Jones.

**Formal analysis:** Elena M. Crecelius, Michael B. Lustik, Milissa U. Jones.

**Investigation:** Elena M. Crecelius, Milissa U. Jones.

**Methodology:** Elena M. Crecelius, Timothy S. Horseman, Milissa U. Jones.

**Project administration:** Timothy S. Horseman, Milissa U. Jones.

**Software:** Michael B. Lustik.

**Supervision:** Milissa U. Jones.

**Visualization:** Milissa U. Jones.

**Writing – original draft:** Elena M. Crecelius, Milissa U. Jones.

**Writing – review & editing:** Elena M. Crecelius, Timothy S. Horseman, Milissa U. Jones.

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
