## [Decision Letter · Decision Letter 0]

18 Feb 2021

PONE-D-21-03174

The prevalence of *M. tuberculosis* among acid fast bacilli cultures from military health system beneficiaries in Hawaii and the Pacific islands from 2002 to 2019

PLOS ONE

Dear Dr. Crecelius,

Thank you for submitting your manuscript to PLOS ONE. After careful consideration, we feel that it has merit but does not fully meet PLOS ONE’s publication criteria as it currently stands. Therefore, we invite you to submit a revised version of the manuscript that addresses the points raised during the review process.

Please submit your revised manuscript. If you will need significantly more time to complete your revisions, please reply to this message or contact the journal office at plosone@plos.org. Please include the following items when submitting your revised manuscript:

We look forward to receiving your revised manuscript.

Kind regards,

Frederick Quinn

Academic Editor

PLOS ONE

Journal Requirements:

2)   Please provide additional details regarding participant consent. In the ethics statement in the Methods and online submission information, please ensure that you have specified (1) whether consent was informed and (2) what type you obtained (for instance, written or verbal, and if verbal, how it was documented and witnessed). If your study included minors, state whether you obtained consent from parents or guardians. If the need for consent was waived by the ethics committee, please include this information.

3)  We note that you have indicated that data from this study are available upon request. PLOS only allows data to be available upon request if there are legal or ethical restrictions on sharing data publicly. For information on unacceptable data access restrictions, please see http://journals.plos.org/plosone/s/data-availability#loc-unacceptable-data-access-restrictions.

Reviewers' comments:

Reviewer's Responses to Questions

**Comments to the Author**

1. Is the manuscript technically sound, and do the data support the conclusions?

Reviewer #1: Yes

Reviewer #2: Yes

2. Has the statistical analysis been performed appropriately and rigorously? 

Reviewer #1: No

Reviewer #2: Yes

3. Have the authors made all data underlying the findings in their manuscript fully available?

Reviewer #1: Yes

Reviewer #2: Yes

4. Is the manuscript presented in an intelligible fashion and written in standard English?

Reviewer #1: Yes

Reviewer #2: Yes

5. Review Comments to the Author

Reviewer #1: See attached word document for full review.

See attached word document for full review.

See attached word document for full review.

See attached word document for full review.

See attached word document for full review.

Reviewer #2: This is a paper that fills a major information gap in the MHS and will be beneficial for future clinical patient care. I was disappointed that the authors did not address the AFBs that were not identified as MTB complex. There are a number of non-TB mycobacteria that can cause clinical illness in humans and are often misdiagnosed and mistreated (such as M.bovis). Since our military service members are often in austere and unusual locations they may have a very different exposure profile than what is seen in the general population. I think the authors missed a chance to determine if that is true. While I would like to see the authors expand the data to determine what the other Mycobacterial species might be, the overall study is straight forward and aligns with current knowledge for the Pacific island region.

6. PLOS authors have the option to publish the peer review history of their article (what does this mean?). If published, this will include your full peer review and any attached files.

Reviewer #1: No

Reviewer #2: **Yes: **Dr. Rebecca L. Pavlicek

---

## [Author Response · Author response to Decision Letter 0]

29 Apr 2021

We appreciate the constructive feedback provided by both reviewers and the opportunity to revise our work in hopes of publication. Below, we address each comment received. 

Reviewer 1:

1. Lack of inclusion of NTM results.

a. We did gather the data on the NTM-positive AFBs and plan to publish those results separately. 

Reviewer 2:

Overall, big picture messages: 

1. VA beneficiaries are not MHS beneficiaries. I understand that TAMC has an agreement with the VA and also care for VA beneficiaries, but they are not the same and VA is not typically included in MHS population denominators for rates.

a. We have separated the results of MHS and VA beneficiaries.

2. Need to ensure that analysis is of individuals rather than specimens because of the necessary assumption of independence. Report tests & positive tests per subject and explain policies & procedures which led to these findings. 

a. Changed results and analyses to be by individual.

3. Prevalence is not incidence or an incidence rate. I recommend obtaining population at risk and calculating actual rates among MHS beneficiaries. This would mean eliminating VA, RC, NG, and PIHCP patients from calculations of rates, as you won’t have denominators for these groups. 

a. Changed calculations to be prevalence based on number of MHS and VA enrollees over the study period, respectively.

b. Included Reserve Corps and National Guard individuals in results and analyses since those RC and NG individuals included in our study are those who were on ‘active’ status at time of sample collection.

c. Removed PIHCP individuals from the analysis as they are a separate demographic group.

4. Recommend further investigation of potential outbreak in 2009 through discussion with public health, including obtaining genotype data to assess previously unsuspected transmission and outbreaks. 

a. The authors have elected to review this data separately and potentially publish a review of all cases of MTB in this sample, Of note, there appears to be no link between any of the cases and no known outbreak of TB during this time. 

5. Recommend shifting discussion from force health protection impact, which is not demonstrated, to testing of low risk populations, which is suggested by these data. 

a. Removed detailed discussion of FHP from discussion section.

Specific comments:

1. Abstract line 29: how does this prevalence of 1% compare to other laboratories in Hawaii, CDC, or in the military? Is it higher or lower? This should be part of the comparison to the literature in the discussion section. I hypothesize that this is a low prevalence of positive cultures but am not sure, which is why comparison data are needed. Then you can further hypothesize why any differences identified exist—likely due to unique policies on TB testing of certain patient types at the Tripler Hospital. I suspect over-testing of low risk populations based on the data provided.

a. Included comparison to Hawaii and United States

2. Line 32: “low rate” –evidence for this is not discussed in the abstract—Prevalence is not a rate. I agree that you should try to ascertain rates so you can make valid comparisons. It is not clear from the data in this manuscript that force health protection measures are effective or making any impact whatsoever. Actual rates of disease would be more compelling. 

a. Discussion now in case rates

3. Line 40 intro: “most” should be quantified. 70% of all US cases are non-US born. Is this higher on Hawaii? 

a. Over 80% born in Asia or US-affiliated Pacific Islands

4. Line 46: MAJOR Problem (potentially) is the inclusion of a different population: VA beneficiaries. These are NOT the same population as MHS beneficiaries and should be excluded from the analysis. This is in contrast with MHS retiree beneficiaries, which is NOT the same thing.

a. Addressed as above

5. Lines 55-57: Not clear what is meant by “unknown true positive rate” or what is being signified here

a. Removed

6. Line 58-59: “high risk” overstates the case. US Service members are a lower risk than the general population, but TB may occasionally occur deployment. Recommend you revise this statement

a. Revised to ‘higher’ risk

7. Line 62: reference 9 is the incorrect reference—should be 8. Please ensure all references are correctly matched. Strongly recommend use of endnote or some other similar software to avoid mistakes like this.

a. This has been corrected. 

8. Line 63: “stringent” probably overstates the case, as does “high risk”. Please revise to accurately convey that screening is meant to test only those at high risk and discourage testing among those at low risk—which is most of the US military population.

a. Revised

9. Line 71-73: Please revise to state something like “…evaluates the prevalence of TB among those MHS beneficiaries who were tested for TB at Tripler Army Medical Center...”

a. Revised

10. Methods line 84: which reference laboratory was used? Hawaii? What were the results? (how did TAMC compare with the reference lab?)

a. Samples were isolated and preliminary identification was performed at TAMC, thereafter samples had to be sent to reference laboratories for further testing. The reference laboratory varied over the study period and was dependent on whichever lab had the contract at the time. That information was not deemed pertinent for inclusion in the study.

11. Lines 89-90: Real problems with identification of MHS beneficiaries evident here. If you’re including NG, then you should also have Reserve Component as well. As above, VA should be considered separately from MHS, as it is completely distinct and separate population. VA are not typically considered part of the MHS beneficiary population (in contrast, retirees who enroll in MHS care are), nor are NG/reserves. I would probably present VA, NG, and RC completely separately as none are typically counted as MHS beneficiaries—what should be the denominator when calculating rates. 

a. Addressed.

12. Lines 99-111: MAJOR issue is the fact that multiple samples should be present for each individual, which means the analysis presented is invalid since independence is an assumption of regression analysis. If you have multiple samples per subject, then your assumption of independence is not met, and your analysis is not valid. I would expect 3 samples for each person who did not have TB (since it takes 3 specimens to rule out TB among TB suspects), and I would expect more for those who did have TB—at least 5 (additional specimens after completion of 2 months of therapy and to test for cure after 5-6 months). This should be clarified and the analysis revised to include only individuals rather than specimens, or some other method of accounting for this dependence. Also the number of specimens per individual should be presented. If it isn’t close to the numbers above, then a detailed explanation of what the policies are at TAMC and why they differ from the established standard of care. 

a. Follow-up for treatment of tuberculosis and subsequent AFB sample collection is performed by Hawaii Department of Health. So samples obtained during initial diagnosis/hospitalization were obtained within our system, however follow-up samples are not available in our system. 

13. Results Line 115: as above, it doesn’t sound like these are individuals, but rather samples/specimens, and this should be presented as individuals. We need to know how many unique individuals had TB out of how many unique individuals were tested. As supplemental information, please describe the number of tests and number of positive tests, including distributions of these numbers per person.

a. Revised to present results by individuals

14. Line 121: data are not typically “stratified” by the outcome, but rather by exposures. Just say when comparing cases and non-cases or something similar.

a. Revised

15. Table 1: use row percents, not column percents—your outcome is prevalence and the rows give that. Column percents are more difficult to interpret. Also, where are VA beneficiaries in this table? Recommend separating out active component as a specific group, and presenting both RC and NG as a separate MHS beneficiary group. 

a. Revised

16. Line 128: why was Fisher’s exact test used for sex specifically? Not clear—this is typically used for cell sizes <5, so I would have expected any of the other variables. 

a. Fisher’s exact test is designed to assess unadjusted associations for 2x2 contingency tables, which is why it was used for gender and not the other variables. We used chi-squared tests to evaluate associations for higher order m x n contingency tables, i.e., for variables with more than two levels. 

17. Line 131 and 135—these numbers are highly suspect to me, since they indicate that >90% of TB suspects had only one sample, when the standard of care is at least 3. Please clarify and explain in detail the policies and procedures at TAMC and how and why they deviate from standard US practice. 

a. Addressed. 

18. Line 142: The larger number of cases in 2009 is worth further examination and investigation. Was this an outbreak? Were these cases identified during contact investigations? Please contact TAMC public health to get further information on whether an outbreak occurred. With their assistance, you can also contact HI state public health department to get genotyping information to look for evidence of transmission and outbreaks which were not identified during contact investigations. 

a. This information is outside the scope of this paper, however could be addressed as part of the follow-up publication.

19. Table 2: Recommend presentation of unadjusted odds ratios as well as a separate column. This will bring into sharp focus the higher prevalence of the PIHCP population; it is not surprising that this association is not significant in adjusted analysis due to collinearity with Asian/PI race/ethnic group. 

a. PIHCP population removed

20. Also you should present the number of observations used in the adjusted analysis so that we can evaluate the impact of missing data. I suspect that around half of the data were used in this analysis?

a. The adjusted analysis used data from all subjects. Race/ethnicity was the only factor with missing data, but instead of excluding subjects with missing race from the analysis we included a separate category of other/unknown. Of 1283 subjects classified as other/unknown race, 927 were reported as other, 274 were reported as null, 76 were reported as unknown, and 6 were reported as Western Hemisphere Indian. 

21. Why did you combine Dependent/retiree/civilian in this table, which is different from table 1? I recommend doing it the same way as table 1. 

a. Civilians are no longer included. Because there were only 4 MHS retirees with a positive TB test, and positivity rates of TB did not differ significantly between retirees and dependents those groups were combined to maximize power and simplify the comparison to be active duty vs. others. 

22. Discussion line 163: MAJOR recommendation—get population at risk and calculate rates, and present this as part of the analysis in the RESULTS section. I like that you have the overall, but you shouldn’t drop it in at the last minute in the discussion section. Presentation of rates (probably you won’t be able to do more than unadjusted rates) overall, by year, and by demographic groups should be possible either by going through TRICARE or by putting in a request to AFHSD. This would complement your analysis by providing compelling data which is not subject to the biases introduced by arbitrary hospital procedures for TB testing. It is also much easier to compare to other state, and military rates. That should go in the results section, and then you can discuss further in the discussion section. However, you will need to think carefully about which groups can be included in your analysis: you will likely have to exclude VA, RC, and NG (and the Pacific Partnership patients) when calculating rates, as they are not typically included in the denominators of MHS beneficiaries. Anyone not included in the denominator should be excluded from the numerator. 

a. Revised.

23. Your discussion section should follow standard format: key results, comparison to the literature, strengths/limitations, judicious interpretation of significance of findings. It should not start with new data not presented in the results.

a. Revised

24. Line 165: “Overestimate”—this should be clarified. If you are including non-MHS beneficiaries in your case counts, then you will get an overestimate of your rates since they are not in the population at risk (denominator). If you exclude those from your analysis of rates (as I recommend), then you’ll have a slight underestimate since you may miss some MHS beneficiary cases which are diagnosed and treated outside of the MHS.

a. Revised

25. Line 175-185: I’m skeptical about these conclusions about FHP. It is not clear that anything mentioned here has had an impact on FHP. In fact, your data suggests that there may have been an unrecognized outbreak in 2009! You also show a higher prevalence of positive cultures in this finding, which is quite odd and inconsistent with good FHP. I suspect when you look at rates, the rates among AD will actually be lower. I recommend that you frame this through the lens of screening procedures at TAMC and how and why they are testing patients there, which I suspect will explain a lot of the differences seen in prevalence of positive cultures. My suspicion is that there is some policy or accepted procedure which results in lots of testing of low risk individuals, particularly retirees. 

a. Addressed.

26. Line 183: I don’t think the evidence supports these statements of “increased risk”—recommend softening this to acknowledge that all available data suggests that military personnel are actually at lower risk.

a. Revised to clarify that we are comparing them to non-traveling persons in the U.S.

27. Line 186: “on the island of Oahu” – not sure where this comes from, and this is the first time this is mentioned. I recommend just sticking with previous language of MHS beneficiaries. Also, don’t use “rates” unless you actually use rates in your analysis—prevalence doesn’t have a time component so it cannot be a rate. 

a. Revised

28. Line 188-190: As above, recommend that you expand upon these unusual assessment procedures and how they may impact your data and the conclusion you draw from them.

a. Revised. 

29. Line 192-3: “effective mitigation strategy” –I’m not convinced that your data indicate or support this statement. I’m more struck that it shows what happens when you overtest a low prevalence population—you get a lot of negatives.

a. Revised

30. Lines 194-205. I don’t find the discussion of homelessness particularly enlightening, and it is highly speculative. Please provide supporting data which show that MHS beneficiaries are at significant risk for homelessness. It may be relevant for VA beneficiaries, but again this is a distinct population which should be analyzed completely distinctly. 

a. Removed

31. Lines 216-7: please assess and acknowledge the effect of collinearity on PIHCP estimates

a. PIHCP removed from study group

32. Line 223: I think this manuscript would have more value if it attempted to elucidate these reasons through examination of policies and procedures, discussion with Public Health regarding contact investigations and outbreaks (including genotyping), and calculation of rates. 

a. This information may be considered for future publications, however was not within the initial intention of this publication. 

33. Line 227: How many of the 11 cases in 2009 were AD? This may have been driving the TB rates in the overall military in 2009.

a. 1 of 7 were active duty. We have included this information in the revised manuscript.

34. Line 240: Typically much more of an issue in dependents rather than military service members, please clarify.

---

## [Decision Letter · Decision Letter 1]

24 May 2021

PONE-D-21-03174R1

The prevalence of *M. tuberculosis* among acid fast bacilli cultures from military health system and Veterans Affairs beneficiaries in Hawaii and the Pacific islands from 2002 to 2019

PLOS ONE

Dear Dr. Crecelius,

Thank you for submitting your manuscript to PLOS ONE. After careful consideration, we feel that it has merit but does not fully meet PLOS ONE’s publication criteria as it currently stands. Therefore, we invite you to submit a revised version of the manuscript that addresses the points raised during the review process.

Please submit your revised manuscript. If you will need significantly more time to complete your revisions, please reply to this message or contact the journal office at plosone@plos.org. Please include the following items when submitting your revised manuscript:

We look forward to receiving your revised manuscript.

Kind regards,

Frederick Quinn

Academic Editor

PLOS ONE

Journal Requirements:

Reviewers' comments:

Reviewer's Responses to Questions

**Comments to the Author**

1. If the authors have adequately addressed your comments raised in a previous round of review and you feel that this manuscript is now acceptable for publication, you may indicate that here to bypass the “Comments to the Author” section, enter your conflict of interest statement in the “Confidential to Editor” section, and submit your "Accept" recommendation.

Reviewer #1: All comments have been addressed

2. Is the manuscript technically sound, and do the data support the conclusions?

Reviewer #1: Partly

3. Has the statistical analysis been performed appropriately and rigorously? 

Reviewer #1: Yes

4. Have the authors made all data underlying the findings in their manuscript fully available?

Reviewer #1: Yes

5. Is the manuscript presented in an intelligible fashion and written in standard English?

Reviewer #1: Yes

6. Review Comments to the Author

Reviewer #1: Thank you for the opportunity to re-review this manuscript. I commend the authors for their professional, thoughtful, and compelling revision. My comments are aimed at providing greater clarity and understanding for readers, and they refer to the line numbers on the “clean” version.

1. Running title probably would better reflect the content as “M. tb among military & veteran beneficiaries in Hawaii” or something similar

2. Abstract line 30: MHS and VA beneficiaries “who were tested” or something similar

3. Intro line 56-61: Typically populations are presented in the results section and belong there, and how you obtained them are in the methods section.

4. Methods section: missing a description of how the population was obtained—ie. describe policies are in place to obtain the specimens. They should be all TB suspects, but please clarify this and any special, additional policies and procedures in place which may lead to additional specimens here. I’m thinking of the “screening” procedures listed later in the manuscript, which need to be clearly articulated. It sounds like this is primarily routine testing with a TST or IGRA on arrival, which would involve clinical evaluation of all positives and thus likely lead to more low-risk specimen collection based on symptom endorsement. Also should be clarified here who gets these screening procedures—all arrivals, all military, only medical personnel, only hospital assigned personnel, etc, as that is not clear.

5. Line 118: military status is variously described as duty status and patient category in the manuscript. Please pick one and stick with it for consistency.

6. Line 121: case rates should be described as incidence rates, as that is the standard terminology, throughout the manuscript.

7. Line 122: As above, need to describe data source for populations (military and VA), who obtained from, etc.

8. Table 1: recommend you call these Total MHS and VA beneficiaries “tested for AFB”

9. Line 159-161: I’m not sure that the comparisons of rates were done correctly—these are calculated differently than standard 2X2 tables. Please confirm you are using comparisons for rates, for example a simple and easy to use one is available at www.openepi.com Please also clarify which population this comparison refers to.

10. Figures 1, 1A and 1B; these are confusing. Figure 1 is fine but is not referenced in the body of the paper that I can find. 1A and 1B are the same figures in the package I received—this needs to be corrected. You could also replace these with a side by side bar graph which makes the comparisons between VA and military easier. Either way, the denominators of person-time used for both military and VA calculations should be explicit somewhere in the manuscript, either below the graph like in Figure 1 or elsewhere.

11. Discussion. Nice and helpful discussion of rates. However, as the title of the manuscript is prevalence among AFB cultures, and most of the analysis focuses on this, please at least add some comments on the key findings of that part of the analysis. In particular, it is worth commenting on the 1% prevalence of positive cultures, whether that was expected, how it compares to other studies, etc.

12. Line 219-221: this assertion needs to be backed up in the results section by some evidence, such as epi investigations or genotyping, then you can interpret it here if you have that. It is not clear what you’re basing this assertion on. Another explanation for the 2009 bump is random variability and small sample sizes.

13. Line 224-228: As above, this needs to be better supported in the methods section with information on local screening policies and procedures. Then you can discuss what the differences are which may affect your results. What about other difference in selection of population—i.e. more Native Hawaiian and Asian population which has a known higher rate of TB. What about increased travel to TB endemic areas in Asia and PI?

14. Line 235-8: “MHS dependents…” as above, please describe these policies and procedures in the methods section.

15. Line 238-9: “Our observations…” This is a dubious claim which doesn’t have much support from the data—again, I’m still unclear what that screening program consists of. I recommend softening it to something more neutral, or stating that further work can be done to evaluate the effectiveness of the screening program.

16. Line 241: awkward wording: I suggest “ universal health care coverage” or something similar

17. Line 241-247” as in comment 15 above, I’m not really convinced about the effect of the screening program. My interpretation is that you have higher rates than most military populations but lower rates than the local/state populations. Your interpretation of the effect of screening should at least take this into account. Overtesting of the MHS population would be expected to result in higher incidence rates by increased, active case finding. In contrast, it would result in a lower proportion of positive specimens—ie. prevalence among AFB specimens.

18. Limitations line 263 and after: to me the most important is selection bias, both in the screening procedures which may have tested low risk suspects, and in the potential for biased numerators and denominators among VA beneficiaries. I disagree with the assessment that the number of VA beneficiaries receiving their care outside military facilities is small (line 270-1)—please provide substantiating data that this is the case in Hawaii, or at least in the VA population in general. All my prior experience with the VA population suggests that this is not the case. If you don’t have data to support, I would characterize the uncertainty in these statement. Similarly, I find the statement in 279-281 dubious, at least for the VA population—I expect that most cases are actually captured elsewhere.

19. Line 266: minor point, but this is really 18 years or 17+.

20. Line 285-6: as above, I think the incidence fits nicely within the context of a higher rate in the local population and a lower rate in the US military population – it is in the middle, which is really what you would expect.

7. PLOS authors have the option to publish the peer review history of their article (what does this mean?). If published, this will include your full peer review and any attached files.

Reviewer #1: **Yes: **James Dominic Mancuso

---

## [Author Response · Author response to Decision Letter 1]

5 Jun 2021

Review Rebuttal Responses regarding:

The prevalence of M. tuberculosis among acid fast bacilli cultures from Military Health System and Veterans Affairs beneficiaries in Hawaii and the Pacific Islands from 2002 to 2019 

We appreciate the constructive feedback provided by the reviewer and the opportunity to revise our work in hopes of publication. Below, we address each comment received. 

1. Running title probably would better reflect the content as “M. tb among military & veteran beneficiaries in Hawaii” or something similar

• Edited

2. Abstract line 30: MHS and VA beneficiaries “who were tested” or something similar

• Edited

3. Intro line 56-61: Typically populations are presented in the results section and belong there, and how you obtained them are in the methods section.

• Edited

4. Methods section: missing a description of how the population was obtained—ie. describe policies are in place to obtain the specimens. They should be all TB suspects, but please clarify this and any special, additional policies and procedures in place which may lead to additional specimens here. I’m thinking of the “screening” procedures listed later in the manuscript, which need to be clearly articulated. It sounds like this is primarily routine testing with a TST or IGRA on arrival, which would involve clinical evaluation of all positives and thus likely lead to more low-risk specimen collection based on symptom endorsement. Also should be clarified here who gets these screening procedures—all arrivals, all military, only medical personnel, only hospital assigned personnel, etc, as that is not clear.

• Added lines 85-87 to clarify how these samples were submitted to lab. 

• These were not screening tuberculosis samples, they were all clinical specimens submitted ordered by physicians. The presumption is that all of these were submitted based on ATS/IDSA/CDC guidelines. To our knowledge none of these samples were submitted during research screening testing. 

5. Line 118: military status is variously described as duty status and patient category in the manuscript. Please pick one and stick with it for consistency. 

• Changed to military status

6. Line 121: case rates should be described as incidence rates, as that is the standard terminology, throughout the manuscript. 

• Changed to incidence rates 

7. Line 122: As above, need to describe data source for populations (military and VA), who obtained from, etc.

• Included in methods (lines 104-106 and 108-110)

8. Table 1: recommend you call these Total MHS and VA beneficiaries “tested for AFB”

• Edited

9. Line 159-161: I’m not sure that the comparisons of rates were done correctly—these are calculated differently than standard 2X2 tables. Please confirm you are using comparisons for rates, for example a simple and easy to use one is available at www.openepi.com Please also clarify which population this comparison refers to.

• Edited, please see lines 170-175

10. Figures 1, 1A and 1B; these are confusing. Figure 1 is fine but is not referenced in the body of the paper that I can find. 1A and 1B are the same figures in the package I received—this needs to be corrected. You could also replace these with a side by side bar graph which makes the comparisons between VA and military easier. Either way, the denominators of person-time used for both military and VA calculations should be explicit somewhere in the manuscript, either below the graph like in Figure 1 or elsewhere.

• The Figure 1 you are referencing was from the initial submission and is not intended for inclusion in the revised manuscript. The file inventory appears to have all prior submissions regardless of intended inclusion in the revision.

• The revised submission had no Figure 1, but had Figure 1A and Figure 1B. Based on your feedback for Figures 1A and 1B, they were consolidated into a new Figure 1 (side-by-side bar graph).

• Added a details regarding how the annual and cumulative incidence rates were calculated in the methods section (lines 127-130). Person-years were not used to calculate the rates. We used the population numbers by year and over the entire study period instead of cumulative person-time as the denominator for our incidence calculations.

• The numerical ranges of enrolled MHS and VA beneficiaries now explicitly stated in lines 161-164. 

11. Discussion. Nice and helpful discussion of rates. However, as the title of the manuscript is prevalence among AFB cultures, and most of the analysis focuses on this, please at least add some comments on the key findings of that part of the analysis. In particular, it is worth commenting on the 1% prevalence of positive cultures, whether that was expected, how it compares to other studies, etc.

• Please see lines 155-156 in results and 218-220 in discussion

12. Line 219-221: this assertion needs to be backed up in the results section by some evidence, such as epi investigations or genotyping, then you can interpret it here if you have that. It is not clear what you’re basing this assertion on. Another explanation for the 2009 bump is random variability and small sample sizes.

• Edited, please see lines 239-240

13. Line 224-228: As above, this needs to be better supported in the methods section with information on local screening policies and procedures. Then you can discuss what the differences are which may affect your results. What about other difference in selection of population—i.e. more Native Hawaiian and Asian population which has a known higher rate of TB. What about increased travel to TB endemic areas in Asia and PI?

• There are many theories that we discussed as being possible causes for the differences in our results, however due to the lack of clear cause of the differences this section was removed. 

14. Line 235-8: “MHS dependents…” as above, please describe these policies and procedures in the methods section.

• Removed this section.

15. Line 238-9: “Our observations…” This is a dubious claim which doesn’t have much support from the data—again, I’m still unclear what that screening program consists of. I recommend softening it to something more neutral, or stating that further work can be done to evaluate the effectiveness of the screening program.

• Removed this section.

16. Line 241: awkward wording: I suggest “ universal health care coverage” or something similar

• Removed this section.

17. Line 241-247” as in comment 15 above, I’m not really convinced about the effect of the screening program. My interpretation is that you have higher rates than most military populations but lower rates than the local/state populations. Your interpretation of the effect of screening should at least take this into account. Overtesting of the MHS population would be expected to result in higher incidence rates by increased, active case finding. In contrast, it would result in a lower proportion of positive specimens—ie. prevalence among AFB specimens.

• Removed this section.

18. Limitations line 263 and after: to me the most important is selection bias, both in the screening procedures which may have tested low risk suspects, and in the potential for biased numerators and denominators among VA beneficiaries. I disagree with the assessment that the number of VA beneficiaries receiving their care outside military facilities is small (line 270-1)—please provide substantiating data that this is the case in Hawaii, or at least in the VA population in general. All my prior experience with the VA population suggests that this is not the case. If you don’t have data to support, I would characterize the uncertainty in these statement. Similarly, I find the statement in 279-281 dubious, at least for the VA population—I expect that most cases are actually captured elsewhere.

• Edited; deleted prior lines 279-281. Addressed previous questions regarding screening procedures; no additional screening procedures using AFB testing are in place. 

19. Line 266: minor point, but this is really 18 years or 17+.

• Edited

20. Line 285-6: as above, I think the incidence fits nicely within the context of a higher rate in the local population and a lower rate in the US military population – it is in the middle, which is really what you would expect.

• Edited wording

---

## [Decision Letter · Decision Letter 2]

24 Jun 2021

The prevalence of *M. tuberculosis* among acid fast bacilli cultures from Military Health System and Veterans Affairs beneficiaries in Hawaii and the Pacific Islands from 2002 to 2019

PONE-D-21-03174R2

Dear Dr. Crecelius,

We’re pleased to inform you that your manuscript has been judged scientifically suitable for publication and will be formally accepted for publication once it meets all outstanding technical requirements.

Kind regards,

Frederick Quinn

Academic Editor

PLOS ONE

Additional Editor Comments (optional):

Reviewers' comments:

Reviewer's Responses to Questions

**Comments to the Author**

1. If the authors have adequately addressed your comments raised in a previous round of review and you feel that this manuscript is now acceptable for publication, you may indicate that here to bypass the “Comments to the Author” section, enter your conflict of interest statement in the “Confidential to Editor” section, and submit your "Accept" recommendation.

Reviewer #1: All comments have been addressed

2. Is the manuscript technically sound, and do the data support the conclusions?

Reviewer #1: Yes

3. Has the statistical analysis been performed appropriately and rigorously? 

Reviewer #1: Yes

4. Have the authors made all data underlying the findings in their manuscript fully available?

Reviewer #1: Yes

5. Is the manuscript presented in an intelligible fashion and written in standard English?

Reviewer #1: Yes

6. Review Comments to the Author

Reviewer #1: I have read through the responses and they seem appropriate. Nice job addressing issues and making corrections. No further comments at this time.

7. PLOS authors have the option to publish the peer review history of their article (what does this mean?). If published, this will include your full peer review and any attached files.

Reviewer #1: No

---

## [Editor Report · Acceptance letter]

29 Jun 2021

PONE-D-21-03174R2 

The prevalence of *M. tuberculosis* among acid fast bacilli cultures from Military Health System and Veterans Affairs beneficiaries in Hawaii and the Pacific Islands from 2002 to 2019 

Dear Dr. Crecelius:

I'm pleased to inform you that your manuscript has been deemed suitable for publication in PLOS ONE. Congratulations! Your manuscript is now with our production department. 

Kind regards, 

on behalf of

Dr. Frederick Quinn 

Academic Editor

PLOS ONE